# Evidence of Adaptation to Increasing Temperatures

**DOI:** 10.3390/ijerph17010097

**Published:** 2019-12-21

**Authors:** Lisbeth Weitensfelder, Hanns Moshammer

**Affiliations:** Department of Environmental Health, Center for Public Health, Medical University of Vienna, Kinderspitalgasse 15, 1090 Vienna, Austria; lisbeth.weitensfelder@meduniwien.ac.at

**Keywords:** climate change adaption, optimal temperature, temperature-related mortality, threshold temperature, time series analysis

## Abstract

In times of rising temperatures, the question arises on how the human body adapts. When assumed that changing climate leads to adaptation, time series analysis should reveal a shift in optimal temperatures. The city of Vienna is especially affected by climate change due to its location in the Alpine Range in Middle Europe. Based on mortality data, we calculated shifts in optimal temperature for a time period of 49 years in Vienna with Poisson regression models. Results show a shift in optimal temperature, with optimal temperature increasing more than average temperature. Hence, results clearly show an adaptation process, with more adaptation to warmer than colder temperatures. Nevertheless, some age groups remain more vulnerable than others and less able to adapt. Further research focusing on vulnerable groups should be encouraged.

## 1. Introduction

Outdoor temperature is an important predictor of overall and cause specific mortality risk. Both cold temperatures [1,2] and hot temperature extremes [3] are associated with higher mortality rates. Therefore, a kind of U-shaped association between daily temperature and daily number of deaths is expected [4,5]. Previous evidence suggests that the optimal temperature, that is the temperature with the lowest number of deaths, is higher in areas with warmer climate and lower in areas with colder climate [6,7]. Assuming that a warming climate leads to an adaptation process, a shift of the local optimal temperature can be expected to higher temperatures.

Like the Polar Regions also the mountainous areas of the earth experience an above average warming process, mostly due to less snow cover leading to a reduced albedo [8]. This also applies for the Alpine Range in Middle Europe [9]. The city of Vienna is located at the Eastern end of that range and is therefore well placed to investigate the hypothesis, that increasing average temperatures are associated with an increase in the optimal temperature.

We have already conducted time series analyses in Vienna before. These previous studies aimed at investigating the health impact of air pollution: Particulate air pollution and nitrogen dioxide in a time series for the years 2000–2004 [10], later also reported in a comparison with other cities [11], and ozone for the years 1991–2009 [12]. In these analyses also the typical U-shaped dose-effect function of temperature on daily mortality was evident, although not in the focus of the study. Temperature then was only included in the models as a possible confounder [13]. Now the temporal change in the temperature-mortality association is examined in more detail and for a longer period (1970–2018). The optimal temperature will be calculated for a series of moving 5-year intervals (1970–1974, 1971–1975, etc.) and a time trend in the optimal temperature will be examined by linear regression over the consecutive 5-year periods.

## 2. Materials and Methods

Mortality data were obtained from the national Austrian Statistics Institute (Statistik Austria). For each death occurring in Austria since 1 January 1970, the following information was provided: Age (in years), sex, date of death, most recent place of residence (district), and primary cause of death. The latter information was provided as International Code of Diagnoses (ICD) version 8 (ICD8) until 1979, as ICD9 until 2001, and as ICD10 from 2002 onwards. After 2015 cause of death was no longer available due to data protection concerns. Because of the changes in diagnostic coding, the lack of cause of death data for the last 4 years, and the lack of specificity of most causes of death for temperature related mortality [14], only total daily mortality was considered, selecting all residents of any of the 23 districts of the city of Vienna. 

Meteorological data (daily mean temperature and daily mean relative humidity) were abstracted from the annual reports of the Austrian Meteorological Service (Zentralanstalt für Meteorologie und Geodynamik, www.zamg.ac.at). The monitoring station is placed at “Hohe Warte” in the Western hilly part of Vienna, outside of the central heat island area.

Mortality data were first examined over time by Excel plots of the raw data and of weekly averages.

All other statistical analyses were performed with STATA 13.1 SE [15]. At first General Additive Models (GAM) were constructed using cubic splines for smoothing of non-linear associations (Family: Poisson) and the resulting curves were plotted to visualize the form of the association. The dependent factor of daily number of deaths was explained by a smooth function of time with 2–4 degrees of freedom per year. In addition, mean same-day temperature and same-day humidity, day of week, and finally also temperature averaged over 2 resp. 4 weeks were included in the model. Previous work has indicated that heat has a very acute effect [16] and that same-day temperature is most predictive of heat-associated deaths. The effect of cold exhibits a much more prolonged effect and the temperature averaged over a longer time period might better capture that impact. 

Upon inspection of the splines simple parametric Poisson regression models were constructed that mimicked the form of the splines. The choice of parameters was guided by the Akaike Information Criterion [17]. The effect of same-day temperature was modeled either with a quadratic term or with a “Hockey stick function”, i.e., as a linear effect above a given threshold level. Then the same model was applied for each moving 5-year interval (1970–1974, 1971–1975, etc.) and the minima of the quadratic function were calculated by setting the derivate of the quadratic function to zero. These minima (“optimal temperature”) were then regressed against years. 

For a “Hockey-stick” model the residuals of the above mentioned “simple parametric Poisson regression model” over the whole period (1970–2018) were calculated. That enabled us to perform linear regression of same day temperature on the residuals which allowed us to use the “nl” command in STATA to define a “Hockey stick function”:nl (residuals = {a} ∗ (t − {b}) ∗ (t > {b}) + {c}),
with {a} signifying the slope of the temperature-response association above a threshold temperature of {b}. Both threshold temperature and slope parameter were then regressed over time. *p* values < 0.05 were considered to be statistically significant.

## 3. Results

### 3.1. Descriptive Data

During the observation period of 49 years (1970–2018; 17,897 days) on average 20,604 deaths per year (1,009,578 in total) occurred. On average 56.4 deaths occurred per day (standard deviation: 15.0, range: 21–138). In spite of a growing population, the daily number of deaths declined over most of the time. This trend was only halted during the final years (Figure 1). 

There was a clear seasonal trend as well in the number of deaths with more people dying in the cold season. This seasonal variation became less pronounced during the latest years when a second peak became more and more prominent during the summer seasons (Figure 2a,b).

The mean daily temperature was 11.0 °C (standard deviation: 8.2 °C, range: −17.3–30.1 °C). Also temperature followed a clear seasonal pattern with cold days in winter and hot days in summer. Over the years a steady increase in annual mean temperatures (0.41 °C per decade, *p* < 0.001) was observed accompanied by an increase in the annual standard deviation (0.1 °C per decade, *p* = 0.041) as well (Figure 3a,b).

### 3.2. GAM Model

General additive Poisson regression models with varying degrees of freedom for time were examined. Relative humidity and average temperature over several weeks (2 or 4 weeks) both exhibited an association with number of deaths that was not significantly different from linearity. Upon inclusion of a term for chronic temperature exposure (2 or 4 week average) same-day temperature exhibited a linear increase of effect above a certain threshold, while in models without chronic temperature the same-day temperature effect clearly resembled a U-shaped curve as expected (Figure 4a,b). 

### 3.3. Parametric Models

Poisson regression models were built that included the following independent parameters: A sine-cosine curve with a wave-length of 365.25 days representing astronomical impacts (e.g., sunshine duration), a linear trend over time (years), same-day relative humidity, dummies for each day of the week, chronic temperature (either previous 14 of 28 day averages), and, finally, parameters for same-day average temperature, modeled as temperature and temperature squared.

In order to find the best fitting averaging time for chronic temperature the Akaike Information Criterion was applied. As averages for the first 14 respectively 28 days of the observation period were missing the various models were applied for the period from 1971 onwards (Table 1). 

The three models (models 1–3 in Table 1) performed equally well (Pseudo R^2^ for all models was 0.26). Akaike’s information criterion was smallest for model 2, i.e., the model controlling for 14-days average temperature. Preliminary analyses (data not shown) had indicated that the coefficients for average temperature (2 or 4 weeks before) did not change systematically over the observation period. Therefore chronic temperature (14 days) according to model 2 but without including same-day temperature was used for the full period (1970–2018). 

Alternatively model 2 (including same-day temperature with a linear and a quadratic term, thus called “quadratic model”) was run over repetitive moving 5-year periods (1970–1974, 1971–1975, and so on). For the same 5-year periods the threshold model was also investigated in more detail. In the quadratic models controlling for chronic temperature effect the minimum of the acute temperature effects were located around 0 °C while in models not including chronic temperature the optimal temperature was between 10 and 20 °C (as is also evident from the spline model, Figure 4b). The threshold model was less affected by the inclusion of chronic temperature. Figure 5a,b depict the results for the temperature with the fewest deaths in the quadratic model (Figure 5a) and for the thresholds (Figure 5b) from models both including 14-days chronic temperature. 

### 3.4. Time Trends in Optimal and in Threshold Temperature

Both “optimal temperature” and “threshold temperature” increased over time (Figure 5a,b). This increase is in the same order of magnitude as the increase in mean temperature: Daily temperature increased by 0.04 °C per year (*p* < 0.001), the optimal temperature increased by 0.09 °C per year, and the threshold for an acute temperature effect increased by 0.07 °C per year. Hence the optimal temperature increased even faster than the average temperature.

The coefficient for the slope above the threshold temperature remained fairly constant around or slightly below 2 for the whole period. The coefficient ranged from 1.3 (for the 5-year-group until 2010) and 2.7 (until 1992). 

### 3.5. Age Risk Groups for High Temperature

Most people die at old age. Slightly more than 50% of all deaths in Vienna occurred at an age above 75 years. Previous analyses [18] have also indicated that a higher relative risk for dying during cold and hot temperatures is also more pronounced in the older age groups. Hence, we additionally did separate calculations for the older age group (75 years and older). Interestingly, this age group displayed a U-shaped curve (similar to Figure 4b) even after adjustment for 14-day average temperature in the GAM investigating the full period (1970–2018). Only during the last 5-year-groups (beginning with the group 2001–2005) was a solution found for the hockey-stick function. Apart from that both models (quadratic and threshold model) displayed nearly the same time trends as in the whole data set (data not shown).

## 4. Discussion

Temperature impacts on health are in the public focus in Austria and Vienna since the 2003 heat-wave [19]. Two first project reports published in 2006 [18,20] investigated the impact of heat-waves and single heat-wave days on daily mortality. In that report a heat-wave was defined following the proposal of Kysely [21]: a period of at least three consecutive days with the maximal temperature exceeding 30 °C. Already this report examined the U-shaped temperature-mortality curve as a back-drop to the additional effect of extreme and prolonged heat. It reported that same day temperature was a better predictor of mortality compared to temperature from previous days. Temperature displayed a strong temporal autocorrelation. Also different temperature parameters (average, maximal, and minimal daily temperature, temperature from “Hohe Warte” and from “Inner city”) were highly correlated with each other. Upon separation of temperature effects in the warm and in the cold season same day temperature was most predictive of hot season effects while a lag of about 5 days had the strongest effect regarding cold effects in winter. Later (unpublished) we observed that the average temperature over a longer time (2–4 weeks) showed even a stronger effect of cold and that it displayed a remarkably linear temperature-effect association.

These previous observations were confirmed by others (e.g., [22]) and fed into the current models. The position of the “optimal” temperature of the “threshold” temperature will certainly depend on various model assumptions. More sophisticated models would consider distributed lags [23,24,25,26] and eventually consider indicators of heat-waves on top of the general temperature effect [19,22]. But with the application of the same model for every 5-year interval the relative change in the optimal temperature and the threshold temperature should be independent from model choices. The results clearly indicate an adaptation response. That should be highly relevant for predicting future impacts of climate change [27].

We do acknowledge that the sine-cosine-function is an uncommon approach, but it enables us to separate astronomical influence (mostly sunshine-duration) from temperature. We also acknowledge that the overall Poisson model is simplified in many ways: The long-term trend was not linear as assumed in the parametric model but a declining trend in daily death counts ended after the year 2000 (Figure 1). For the quadratic model this posed no problem because different coefficients for the long-term trend were used for each 5-year group, but the residuals used in the hockey-stick model were systematically biased for the latest years. However, this systematic shift in numbers of death unexplained by seasonal and long-term trends could not affect estimates of acute temperature effects. The additional peaks in death counts in summer in the latest years (Figure 2b) are not interpreted as a sign that the sine function for astronomical influences is no longer valid. A more plausible explanation would be the increasingly hot summers with prolonged heat-waves leading to these additional peaks in mortality. Modeling temporal trends as splines would have covered also these likely temperature-associated effects rendering a spline-model over-adjusted for the study of acute and delayed temperature effects. Generally our Poisson model was simplified on purpose. To calculate precise cut-offs or optimal temperatures parametric models were necessary. We found it more appropriate to use a fully parametric model instead of a mixture between parametric functions and splines. However, comparisons with the spline model and a negative binomial regression model (data not presented) indicate very similar results and hence not a severe over-dispersion in the Poisson model. Therefore, we are confident that the change in the point estimates (optimal temperature and threshold temperature) over time is estimated fairly accurately. From the data presented in Figure 5
*p*-values cannot be calculated directly because of overlapping time periods. Therefore we constructed five series each of non-overlapping time periods. For the optimal temperature we found an increase of 1.3–2.5 °C per decade with *p*-values ranging between 0.001 and 0.074 (only one *p* > 0.05). For the threshold temperature we found increases of 0.5 and 0.9 °C per decade with *p*-values ranging between 0.001 and 0.125 (2 *p*-values > 0.05).

Interestingly, in the older age groups the effect of acute cold temperature was still evident even after controlling for chronic (14-day average) cold effects. This might indicate that older people are more acutely affected by cold spells than younger people. However, certainly more research into the adaptation potential of different risk groups (by age, sex, or underlying diagnoses) is warranted. Nevertheless, we report clear signs of adaptation to climate change both in the total population of Vienna and in the elderly (75 years and more).

Adaptation is due to different mechanisms: physiological adaptation (circulatory system, sweat production, etc.), behavioral adaptation (clothing, drinking, food uptake, outdoor activities, etc.), cultural adaptation (organization of daily distribution of work and rest, etc.), and constructive adaptation (housing, insulation, infrastructure, etc.), but on the population level adaptation can also be a sign of selection: in a first heat-wave susceptible persons are more likely to die. In a second heat-wave, the pool of susceptible persons will necessarily be smaller. Therefore, adaptation processes occur at different time scales. While the fastest are seen within one single season [16] other responses show mortality displacement across seasons [28,29] or might even take years [30].

Other factors will affect heat (and cold) impact as well, as for example air pollution levels [31]. While adaptation to a general shift in temperature distribution will likely occur there is also evidence that climate change will also increase the variability of weather parameters as also demonstrated for temperature in the observation period. Increasing variation means an increase in intensity and frequency of extreme events and for these an adaptation response is much less likely to occur. Various individual and population level determinants of vulnerability must also be taken into account [22]. Individual determinants of vulnerability will not only determine current impacts of extreme temperatures but even more so future trends and the ability to adaptation.

## 5. Conclusions

We found signs of adaptation to increasing temperatures. The temperatures with the lowest numbers of daily deaths (“optimal temperature”) increased over time and this increase was even faster than the increase in daily temperature. Also, the threshold in same day temperature above which the number of daily deaths increases increased over time. The estimates of that threshold were nevertheless not very precise and varied considerably over time. 

It is still not clear which mechanisms were driving adaptation to temperature rise in a generally temperate climate. Further research should focus on the adaptation capacity of vulnerable groups and the limits to adaptation especially in view of increasing temperature variability.

## Figures and Tables

**Figure 1 ijerph-17-00097-f001:**
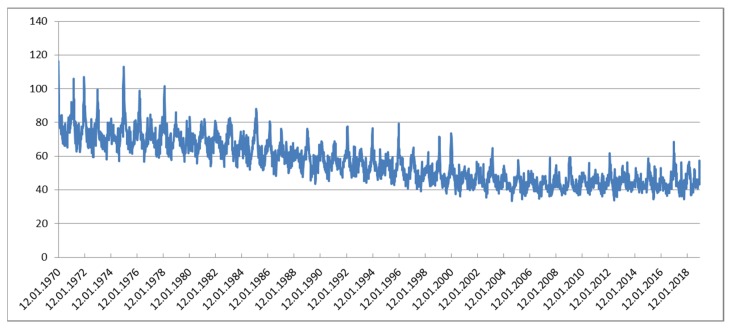
Number of daily deaths (weekly averages). From 1970 until after the year 2000 a declining trend is evident. Numbers of deaths peak each winter with some higher peaks most likely because of severe influenza epidemics.

**Figure 2 ijerph-17-00097-f002:**
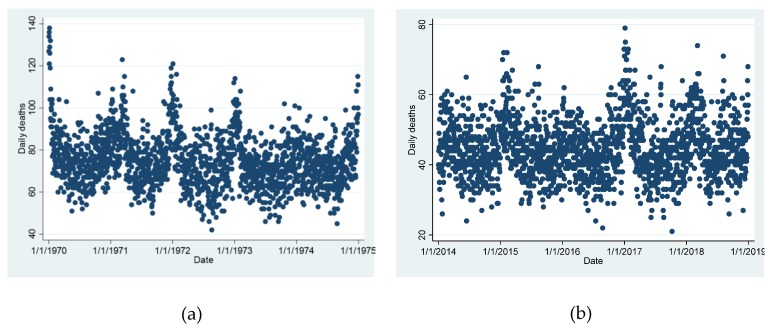
Number of daily deaths during selected 5-year periods: (**a**) the first 5 years, with a clear seasonal pattern; (**b**) the final 5 years, where the seasonal pattern is interrupted by additional peaks in summer.

**Figure 3 ijerph-17-00097-f003:**
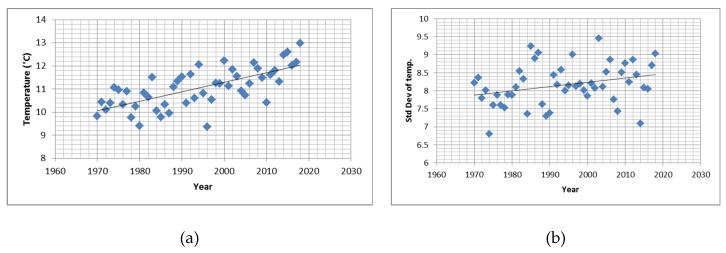
Time-trends in temperature: (**a**) annual means; (**b**) standard deviation.

**Figure 4 ijerph-17-00097-f004:**
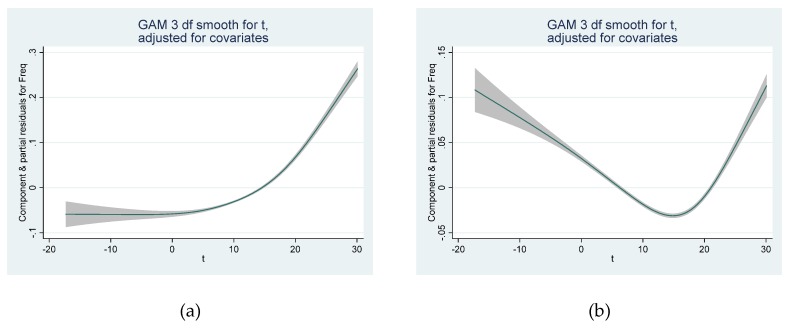
Shape of the dose-response function (spline with 3 degrees of freedom) between same-day temperatures (°C); (**a**) in a model including average temperature over the previous 28 days, and (**b**) models not including chronic temperature effects.

**Figure 5 ijerph-17-00097-f005:**
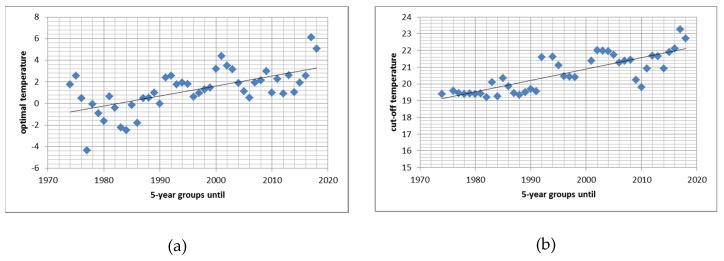
Effect of adaptation to rising temperatures: Increase in optimal temperature (**a**) and in threshold temperature (**b**) calculated for 5-year periods.

**Table 1 ijerph-17-00097-t001:** Comparison of different parametric models.

**Model 1**	**Coefficient**	***p*-Value**	**R^2^ = 0.26**	**AIC = 125,306.1**
sine	0.025	<0.001		
cosine	0.074	<0.001		
Year	−0.014	<0.001		
Tue-Sun^1^	−0.041; 0.004	<0.001; 0.89		
Temp	−0.0015	<0.001		
Temp-squared	0.0005	<0.001		
Rel. Humidity	0.0001	0.138		
28-day temp.	−0.010	<0.001		
Constant	32.04	<0.001		
**Model 2**	**Coefficient**	***p*-Value**	**R^2^ = 0.26**	**AIC = 125,238.9**
sine	0.042	<0.001		
cosine	0.089	<0.001		
Year	−0.018	<0.001		
Tue-Sun^1^	−0.040; 0.004	<0.001; 0.88		
Temp	−0.0001	0.882		
Temp-squared	0.0004	<0.001		
Rel. Humidity	0.0002	0.066		
14-day temp.	−0.0097	<0.001		
Constant	32.18	<0.001		
**Model 3**	**Coefficient**	***p*-Value**	**R^2^ = 0.26**	**AIC = 125,643**
sine	0.069	<0.001		
cosine	0.150	<0.001		
Year	−0.014	<0.001		
Tue-Sun ^1^	−0.041; 0.004	<0.001; 0.90		
Temp	−0.0038	<0.001		
Temp-squared	0.0005	<0.001		
Rel. Humidity	0.0002	0.017		
Constant	32.61	<0.001		

^1^ Compared to Mondays; Tuesdays and Wednesdays have a positive coefficient. Only Saturdays and Sundays differ significantly from Mondays (all models).

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
