# Peer review of "Evidence of Adaptation to Increasing Temperatures"

_ijerph, 2019, doi:10.3390/ijerph17010097_

Round 1

Reviewer 1 Report

This is an interesting paper trying to find evidence of adaptation to increasing temperature from a time-series analysis of daily mortality and temperature data in Wien.

Overall the study is well conducted and the analytic methods coherent with the research question.

I have major issues that I would rise:

My understanding is that the authors included in the model sine, and cosine pairs, along with calendar year to control for seasonal and long terms trends. With this parametrisation they are assuming that the seasonal pattern is the same across the years, while they are giving empirical evidence (Figure 2) that the seasonal pattern is changing over time. In the last years there has been a rising consensus on using a more aggressive control for seasonal and long terms trends, e.g. using smoothing splines with 7 degree of freedom per year (not assuming a constant seasonal pattern). All the adaptation mechanism is investigating through the minimum mortality temperature (MMT). What about the RR? Are these changing over time. Figure 2 again suggests that people looks more vulnerable during summer during the last years. I’m not a fan on estimating parameters (e.g. MMT) over repeated moving-average period. This gives a clearer patter of the derived trends (e.g. Figure 5), but it produces autocorrelation on the yearly time-series, and the authors should consider of the induced auto-correlation in order to avoid an inflated precision on the estimates.

Other minor points:

Introduction, lines 23-24: I would change the sentence “Therefore must be a kind of U-shaped”. Material and Methods, line 62: In some time-series analysis is not unusual to fine over-dispersion on the mortality counts. Assuming a Poisson distribution could induce an inflated precision on the estimates. Have the authors estimated the amount of over-dispersion in their data? Results, line 79: 17,897 deaths in 49 years looks a small number. Is it correct? Results, lines 87-89: The increase of mean temperature and SD is significant? Results, parametric models lines 108-139. I think some of the content of this paragraph should go in the method section. Surely the description of the “Hockey stick function” model.

Author Response

Thank you for the detailed review! Please find our detailed response in the attached document.

Reviewer 2 Report

The paper describes long-term changes of optimum temperature in Vienna with the increase of temperature. The results are interesting and valuable. However, I would like to make some confirmation about the procedure of analysis.

The major concern of the reviewer is whether the degree of freedom was properly given in assessing the statistical significance (p-values) of results. Statistics based on moving windows results in a reduced degree of freedom, which is one-fifth of that of the original time series if a five-year window is used (1970-74, 1971-75, ---). I would like to make sure that the reduction of degree of freedom was taken into account in the analysis.

In addition, I failed to understand the exact procedure of calculating the optimal and threshold temperatures. The graph in Fig.4b shows a well-defined minimum at around 15C. Was the optimal temperature defined as the point of minimum in this kind of graph? How was the threshold temperature calculated?

Author Response

Thank you for the suggestions. Please find our detailed response in the attached file!

Round 2

Reviewer 1 Report

I think the authors respond positively to my previous comments.

I have the following questions/comments:

1) Introduction line 24 I would change "must be expected" with "is expected".

2) The interpretation of sine-cosine pairs in terms of astronomical influence (sunshine duration) is very naïve. Seasonality on mortality is a well known phenomena. I would change line 117 in the methods, and lines 193-196 in the discussion.

3) The use of a single spline function of time is not just to directly model seasonality, but to filter out potential confounding by unknown factors varying at a timescales different than daily variation (see for instance Zeger et al 2006, Peng et al JRSSA 2006, Touloumi et at Statistics in Medicine 2006). This method has been applied in all the major multi-city time series investigations in the last 15 years. In discussion the sentence (lines 196-198) "Modelling temporal trends as splines..." goes against the evidence provided by several authors in the last 2 decades.

Author Response

please see our response in the word file!

Reviewer 2 Report

I appreciate the authors' effort of revision. The manuscript is ready for acceptance.

I think "301C" in Line 92 (Page 3) is mistyped.

Author Response

Thank you. We changed the text accordingly.